# Novel Weight-Based Approach for Soil Moisture Content Estimation via Synthetic Aperture Radar, Multispectral and Thermal Infrared Data Fusion

**DOI:** 10.3390/s21103457

**Published:** 2021-05-15

**Authors:** Oualid Yahia, Raffaella Guida, Pasquale Iervolino

**Affiliations:** 1Centre des Techniques Spatiales, Algerian Space Agency, Arzew 31200, Algeria; 2Surrey Space Centre, University of Surrey, Guildford GU2 7XH, UK; r.guida@surrey.ac.uk; 3Airbus Defence and Space, Connected Intelligence, Guildford GU2 7AG, UK; pasquale.iervolino@airbus.com

**Keywords:** soil moisture content, data fusion, integral equation model, Sentinel-1, perpendicular drought index, temperature vegetation dryness index, Landsat-8, feature level fusion, artificial neural network, decision level fusion

## Abstract

Though current remote sensing technologies, especially synthetic aperture radars (SARs), exhibit huge potential for soil moisture content (SMC) retrievals, such technologies also present several performance disadvantages. This study explored the merits of proposing a novel data fusion methodology (partly decision level and partly feature level) for SMC estimation. Initially, individual estimations were derived from three distinct methods: the inversion of an Empirically Adapted Integral Equation Model (EA-IEM) applied to SAR data, the Perpendicular Drought Index (PDI), and the Temperature Vegetation Dryness Index (TVDI) determined from Landsat-8 data. Subsequently, three feature level fusions were performed to produce three different novel salient feature combinations where said features were extracted from each of the previously mentioned methods to be the input of an artificial neural network (ANN). The latter underwent a modification of its performance function, more specifically from absolute error to root mean square error (RMSE). Eventually, all SMC estimations, including the feature level fusion estimation, were fused at the decision level through a novel weight-based estimation. The performance of the proposed system was analysed and validated by measurements collected from three study areas, an agricultural field in Blackwell farms, Guildford, United Kingdom, and two different agricultural fields in Sidi Rached, Tipasa, Algeria. Those measurements contained SMC levels and surface roughness profiles. The proposed SMC estimation system yielded stronger correlations and lower RMSE values than any of the considered SMC estimation methods in the order of 0.38%, 1.4%, and 1.09% for the Blackwell farms, Sidi Rached 1, and Sidi Rached 2 datasets, respectively.

## 1. Introduction

A comprehensive understanding of a variety of hydrological processes and agricultural applications requires accurate surface soil moisture content (SMC) level estimations. SMC has a key role in global applications such as climate change studies, as well as contributing to the determination of a variety of land-atmosphere interactions [1,2]. SMC is also a key factor for medium-to-small level applications such as natural resources management, drought assessments [3], and, especially, agricultural practices like irrigation scheduling [4].

Given the inherent importance of SMC, a multitude of investigations have been pursued in the literature to design retrieval methodologies using different sensing platforms [5,6]. Direct in situ measurements of SMC provide the most accurate estimations, but that accuracy is expensive in terms of time and effort, especially considering that discrete measurements are point-based, which makes them exclusively specific to particular locations and does not offer a realistic depiction of the spatial distribution and variability of soil moisture [6]. These limitations can be overcome by the use of remote sensing [7].

Remote sensing is capable of offering the continuous spatial and temporal coverage of SMC at all levels, and mission-purposed satellites like Soil Moisture And Ocean Salinity (SMOS) [8] and Soil Moisture Active Passive (SMAP) [9] are excellent examples of such coverage [10]. Indeed, such satellites provide accurate SMC estimations (4% error) at a depth of 0–5 cm every three days [11]. However, their respective spatial resolutions (30–50 km for SMOS and 10–40 km for SMAP) render them impractical at the regional level [12]. Conversely, small scale agriculture or family farms (<2 ha), which constitute 75% of the agricultural land of the world [13], require a different set of sensors with a significantly better spatial resolution, namely synthetic aperture radars (SAR) [14], thermal infrared imagers, and multispectral imagers [15].

High-resolution SAR imagers possess night and day imaging capability, are independent of weather conditions, and have a surface penetration ability that varies from a few centimetres (for sensors operating at the X-band) to tens of centimetres (for sensors at the L-band) in dry soil conditions [16]. The backscattered radar signal is directly influenced by the soil moisture content via the dielectric constant of the soil and is inherently sensitive to the soil roughness and texture [17]. The relationship of the dielectric constant and SMC has been described as a polynomial [18]. Numerous models of different natures successfully and consistently use this relationship, whether they are semi-empirical like those of Oh [19] and Dubois [20] or theoretical like the Integral Equation Model (IEM) [21,22]. The IEM has been exhaustively used for the retrieval of soil moisture content and surface roughness parameters [23]. However, it exhibits a questionable performance when the areas of interest are characterized by medium-to-intense vegetation cover, which is responsible for a pronounced reduction in radar response sensitivity to SMC in such areas, especially at very short radar wavelengths [24]. On the other hand, SMC estimations using multispectral and thermal infrared synergies are not the least affected by the presence of partial or even full vegetation cover [25]. The designs of these synergies are based on the strong correlation between surface radiant temperatures and the distribution, as well as the variability of SMC levels and vegetation [26]. Therefore, surface radiant temperatures are often represented by land surface temperature (LST) and vegetation cover intensity [27]. LST is derived from atmospherically corrected thermal infrared images (with wavelengths ranging from 8 to 13 microns), whereas vegetation cover intensity can be represented by a variety of vegetation indices (VI), which are often derived from algebraic combinations of the visible red (380–760 nm) and near-infrared (760 nm^−1^ microns) [28]. The authors of [29] suggested that the relationship between LST and VI may offer an indication of SMC levels. When VI/LST data points are plotted in a two-dimensional scatter plot, a triangular/trapezoidal feature space is formed. The latter is key for the determination of the extreme boundaries (dry/wet edges) necessary for the calculation of an index called the Temperature Vegetation Dryness Index (TVDI) that is characterized by a linear relationship with SMC [30]. However, in the case of variable atmospheric conditions, the performance of the TVDI may be at risk of uncertainty and subjectivity, not to mention the limitations imposed by the shortcomings of current satellite technology used in this synergy, especially coarse temporal resolution and susceptibility to cloudy conditions [27].

Since all sensors and methods discussed above produce results with variable degrees of success under different conditions, data fusion techniques have been proposed as a potential solution. For optimal SMC estimations in terms of accuracy, information is extracted from multiple sensors and merged instead of inferring SMC estimations from a single sensor [31]. The goal of this research work was to address the performance issues of the above-highlighted SMC estimation methods by designing a system with reconfigurable capabilities for SMC retrieval through the exploitation of multiple EO sources and methods. The aim was to design a new system that incorporates a multi-level data fusion to achieve an SMC determination with optimal accuracy. The proposed system ensures the retrieval of SMC levels using satellite data via:The inversion of an updated version of the IEM.The use of a multispectral index called the Perpendicular Drought Index (PDI).The exploitation of the TVDI using a synergy of thermal and multispectral images.A feature level fusion based on different combinations of features.A decision level fusion where all achieved estimations are considered in a weight-based system.

The eventual goal was to achieve better accuracy of SMC estimation with a fusion scheme than that of each method separately.

The paper is organized as follows: Section 2 illustrates the data and proposed methodologies, Section 3 provides a detailed description of the study areas, and Section 4 offers an analysis of the achieved results. Lastly, the research work is completed with conclusions.

## 2. Methodology

This section offers a detailed view of each of the indices and models involved in the data fusion scheme, namely the inversion of IEM, the TVDI, the PDI, the proposed feature level fusion, and, finally, the novel architecture for SMC estimation based on a decision level fusion scheme of the methods above.

### 2.1. The Considered Integral Equation Model

SAR is a popular active microwave technology with relevant potential for SMC and surface roughness parameter retrieval at regional scales [32,33]. In addition to SMC levels, a variety of other factors contribute to the backscattered radar signal, from surface characteristics like the surface roughness profile, the mineralogical composition of the soil, and the dielectric features of the soil to radar characteristics like the incidence angle (from nadir), the operating frequency of the radar sensor, and polarisation [24]. Moreover, the soil dielectric features or dielectric constant (*εs*) exhibit a polynomial relationship with SMC, and other dependencies have been found between SMC and the mineralogical composition of soil and SAR frequency [18,34].

The soil moisture estimation performed in this study was based on the inversion of the single scattering contribution to the radar cross-section as modelled by the IEM [23].

The IEM is a theoretical model for backscattered radar signal representation with an established ability in estimating SMC and surface roughness parameters, as demonstrated in numerous studies [35,36,37,38,39]. The IEM has a wide validity range for a variety of surface roughness values commonly found in agricultural surfaces (satisfying the *k·s* ≤ 3 condition, where *k* is the wavenumber and *s* is the root mean square (RMS) of surface heights) [40].

Equations (1)–(9) represent the backscattering coefficient of the surface contribution [23]:(1)$$\sigma_{pp}^{0}=\frac{k^{2}}{2}e^{\left(-2k_{z\;}^{2}s^{2}\right)}\overset{\infty }{\underset{n=1}{\sum }}s^{2n}{\left|I_{pp}^{n}\right|}^{2}\frac{W^{n}\left(-2k_{x},0\right)}{n!}$$
(2)$$I_{pp}^{n}={\left(2k_{z}\right)}^{n}f_{pp}\;exp\left[-2k_{z}^{2}s^{2}\right]+\frac{1}{2}\left\{k_{z}^{n}\left[F_{pp}\left(-k_{x},0\right)+F_{pp}\left(k_{x},0\right)\right]\right\}$$
(3)$$f_{vv}=\frac{2R_{v}}{\cos\theta_{i}}$$
(4)$$f_{hh}=\frac{-2R_{h}}{\cos\theta_{i}}$$
(5)$$F_{vv}\left(-k_{x},0\right)+F_{vv}\left(k_{x},0\right)=\frac{2\sin^{2}\theta_{i}{\left(1+R_{v}\right)}^{2}}{\cos\theta_{i}}\left[\left(1-\frac{1}{\varepsilon_{s}}\right)+\frac{\mu_{r}\varepsilon_{s}-\sin^{2}\theta_{i}-\varepsilon_{s}\cos^{2}\theta_{i}}{\varepsilon_{s}^{2}\cos^{2}\theta_{i}}\right]$$
(6)$$F_{hh}\left(-k_{x},0\right)+F_{hh}\left(k_{x},0\right)=\frac{2\sin^{2}\theta_{i}{\left(1+R_{h}\right)}^{2}}{\cos\theta_{i}}\left[\left(1-\frac{1}{\varepsilon_{s}}\right)+\frac{\mu_{r}\varepsilon_{s}-\sin^{2}\theta_{i}-\mu_{r}\cos^{2}\theta_{i}}{\mu_{r}^{2}\cos^{2}\theta_{i}}\right]$$
(7)$$W^{\left(n\right)}\left(a,b\right)=\frac{1}{2\pi }\iint \rho^{n}\left(x,y\right)e^{-i\left(ax+by\right)}dxdy$$
where $\sigma_{pp}^{0}$ is the backscattering coefficient, with *pp* signifying the polarisation state (co-polarized SAR data are then used); *θ_i_* is the incident angle; *k_z_ = kcosθ_i_*; *k_x_ = ksinθ_i_*; $R_{h}$ and $R_{v}$ are the horizontally and vertically polarized Fresnel reflection coefficients, respectively; $\varepsilon_{s}$ and $\mu_{r}$ are the relative permittivity and permeability, respectively, of the soil; and $W^{n}$ is the Fourier transform of the *n*th power of the surface correlation function $\rho \left(x,y\right)$. The latter presents an exponential distribution (Equation (8)) for low surface roughness values and a Gaussian (Equation (9)) for high surface roughness values [1]. For one-dimensional surface roughness profiles, the correlation functions are expressed in Equations (8) and (9):(8)$$\rho \left(x\right)=e^{-\frac{\left|x\right|}{l}}$$
(9)$$\rho \left(x\right)=e^{-\frac{x^{2}}{l^{2}}}$$
where *l* is the correlation length.

The roughness of a surface influences the backscattered signal from that surface, justifying the importance of the correct identification of a model for the surface roughness profile [2] and level of roughness. The Rayleigh criterion is frequently used to establish the degree of smoothness of a given surface. A surface is considered rough if it satisfies Equation (10):(10)$$s>\frac{\lambda }{8cos\theta }$$
where λ is the wavelength of the incoming electromagnetic radiation [3].

The RMS height can be calculated using Equations (11) and (12):(11)$$s=\sqrt{\frac{1}{N}\left[\left(\overset{N}{\underset{i=1}{\sum }}Z_{i}^{2}\right)-N\;{\bar{Z}}^{2}\right]}$$
where
(12)$$\bar{Z}=\frac{1}{N}\overset{N}{\underset{i=1}{\sum }}Z_{i}$$
where *N* is the number of points and $\bar{Z}$ is the mean of heights [4].

As for the approximation of the correlation length *l*, it is obtained once the normalised autocorrelation function $\rho \left(\xi \right)$ is defined. The latter is calculated using Equation (13):(13)$$\rho \left(\xi \right)=\frac{\sum_{i=1}^{N-j}Z_{i\;}Z_{i+j}}{\sum_{i=1}^{N}Z_{i}^{2}}$$

The surface correlation length *l* is the horizontal distance over which the surface profile is auto-correlated with a value larger than *1/e* [5].

Due to the mathematical complexity of IEM, alternative methods are used to invert it to calculate SMC, mostly commonly artificial neural networks (ANNs) [6]. ANNs have an established ability to invert IEM for SMC and roughness parameter retrieval, as investigated in numerous studies [1,7,8,9]. An ANN is a parallel distributed information processing structure composed of processing elements interconnected together with unidirectional signal channels named connections or weights [10]. The ANN used for the IEM inversion is a multi-layer perceptron (MLP) [6]. An MLP consists of an input layer, one or many hidden layers, and an output layer, and the training algorithm uses ground truth data to minimise error [10,11]. The IEM inversion does suffer from some performance issues due to several factors summarised by the following bullet points:Speckle is an interference characterised as multiplicative noise typical of coherent sensors [12]. Consequently, such interference leads to a granular appearance of SAR images [13]. Filters can consequently be applied to obtain better soil moisture content retrieval, which comes at the expense of soil moisture heterogeneity in the filtered pixels [13].The effect of SMC on radar signals is less pronounced when SMC levels exceed 35%, especially at the HH polarisation [14].The dielectric behaviour of the soil is substantially affected by the distribution of grain size. The latter determines the amount of free space for available water within the soil, which emphasises the significance of the accurate identification of the mineralogical composition of that soil [15].The accuracy of IEM-based SMC retrieval is largely influenced by the characterisation and accuracy of the measurement of surface roughness parameters [5]. However, the authors of [16,17] proposed semi-empirical calibrations of RMS height and correlation length to improve the characterisation of surface roughness parameters, which produced better results.The revisit time of high-resolution SARs is still inadequate, which makes tracking SMC temporal variations difficult [18].Susceptibility to medium-to-thick vegetation cover, which consequently causes volume scattering, has a direct and negative impact on the accuracy of SMC retrieval [19].Due to the limitations described above, alternative methods to estimate SMC using a different group of sensors, namely multispectral and thermal remote sensors, were explored.

As will become clearer later, for one of the areas in the case study, the measurements collected to estimate the surface roughness parameters showed that the condition (*k·s* ≤ 3) was not satisfied, which means the IEM, as described above, could not be applied in principle. Therefore, the following measures were necessary to increase the IEM validity range:The inclusion of a semi-empirical calibration parameter *Lopt* [20].The implementation of an updated version of the IEM called the Empirically Adapted Integral Equation (EA-IEM) in [21].

### 2.2. The Updated Version of the IEM

In [20], the authors introduced a semi-empirical calibration of IEM by replacing the estimation of correlation length *l* with a calibration parameter called *Lopt*. It was done due to the fact the correlation length is the most difficult parameter to measure, especially if the used profilometers are 1–2 m long (which produces an error of 50%) [22]. Therefore, *Lopt* was suggested as a forcing parameter that accounts for both a better approximation of *l* and any inaccuracies of the IEM model. Furthermore, plotting the IEM as a function of *l* for a given configuration, specifically, a configuration consisting of frequency (s), SMC, polarisation, and incidence angle, revealed that its corresponding radar measured backscattering coefficient had two possible solutions, *Lopt1* and *Lopt2*, and both ensured a good agreement between backscattering coefficients produced by the radar and IEM model. The results of the fitting process suggested that *Lopt2* ensures a more correct representation of the physical behaviour of the backscattering coefficient as a function of *s*, especially if the correlation function is Gaussian [17]. Since all the correlation functions related to this study were Gaussian, the coefficients of *Lopt2* were considered and are referred to as *Lopt* henceforth.

Moreover, no further modifications of the IEM model were necessary, which made this a semi-empirical calibration. For a given frequency, *Lopt* is dependent on the RMS of the surface heights, incidence angle, and polarisation, and the parameters relevant to the configuration of this research work are expressed by Equation (14) [8]:(14)$$Lopt\left(s,\theta_{i},VV\right)=1.281+0.134{\left(sin0.19\theta_{i}\right)}^{-1.59}s$$
where *s* is the RMS of surface heights, $\theta_{i}$ is the incidence angle at pixel *i*, and *VV* is the polarisation.

*Lopt* demonstrates a better agreement between radar signal calculated from the IEM and SAR data at the C-band (at 5.6 cm wavelength) in both HH and VV polarisation, as well as incidence angles from 20° to 48° with an improved validity range of (*s* < 4 cm) [17], which can produce a more accurate IEM inversion with a decreased bias and root mean square error in terms of SMC estimation.

This version of the IEM [23] is used independently from ground truth measurements. The IEM is calculated for a specific range of the parameters discussed above, then the results are fed to an ANN. The resulting network is trained on simulations then validated by ground truth measurements. However, in this research, the authors proposed using the EA-IEM [21] to directly infer the dielectric constant from remotely sensed SAR data instead of using the empirical model of [24]. This work focused especially on the equations related to VV-polarisation since this was the only co-polarized configuration available in the Sentinel-1 datasets. The idea was to directly infer the dielectric constant from the active microwave backscattering coefficient knowing that [21]:(15)$$F_{v}=\frac{\sigma_{vv}^{0}}{\frac{k^{2}}{2}\;exp\left(-2k_{z}^{2}s^{2}\right)\sum_{n=1}^{\infty }\frac{{\left(2sk_{z}\right)}^{2n}W^{n}\left(-2k_{x},0\right)}{n!}}$$
(16)$$\varepsilon_{r}=\frac{1}{{\left(0.5-{\left(\frac{F_{v}s^{0.05}sin^{3.35}\left(\theta +1.1\right){\left(Lopt-0.049\right)}^{\left[0.042+0.06sin\left(\theta -1\right)\right]}}{106exp\left(-1.996s^{2}k_{z}^{2}\right)tan^{0.46}\left(\theta +0.32\right)}\right)}^{\frac{5}{27}}\right)}^{\frac{1}{\cos\left(1.02\theta -0.2\right)}}}-3$$
where *F_v_* in Equation (15) is calculated using the calibrated $\sigma_{vv}^{0}$ extracted from Sentinel-l and the dielectric constant $\varepsilon_{r}$ in Equation (16) is calculated for the Gaussian surface correlation function due to the rough nature of the surface height measurements, where the correlation length was replaced by *Lopt*.

Despite the measures described above, SMC estimation through the inversion of the EA-IEM still has no parameters to account for the vegetation effect and high SMC levels (>35%) on the radar signal. These are the reasons why the authors of this study chose to add relevant parameters suitable for this particular purpose. The parameters in question are the PDI and the TVDI.

### 2.3. Perpendicular Drought Index

Incident radiances in the violet, blue, and red wavelengths are potently absorbed by vegetation lamina tissues, whereas near-infrared (NIR) wavelengths are highly reflected. High vegetation cover intensity signifies small reflectance in the red band and high reflectance in the NIR bands [25]. Since the absorption of the red range is rapidly saturated, the increase of vegetation cover intensity can only be reflected by the increase of the reflectance in the NIR region. The reflectance of bare soil is typically high in the red-to-NIR spectral region; however, the presence of water content in bare soil results in a decrease in said reflectance, especially in the NIR domain [26]. Numerous investigations have found that plotting atmospherically corrected red band pixels against their NIR counterparts results in a triangular spectral feature space that can represent vegetation cover and SMC conditions [25,27,28], as depicted in Figure 1.

The soil line (bare soil) can be expressed using Equation (17) [26]:(17)$$R_{NIR}=MR_{red}+I$$
where *R_red_* and *R_NIR_* are surface reflectance derived from red and NIR bands, respectively, and M and I are, respectively, the slope and intercept of the soil line in the NIR–red feature space [28].

In [26], it was suggested that any mathematical operation reinforcing the contrasts between NIR and red could be used to indicate the vegetation surface drought status and discriminate bare soil pixel information from that of vegetated pixels. In the same paper, the concept of an orthogonal axes system, above the aforementioned triangular feature space and represented by an index named the *PDI*, was proposed. Indeed, the observation of Figure 2 is necessary to fully comprehend the concept of the *PDI*.

Figure 2 illustrates the concept of PDI where:AD line is a representation of the change in terms of vegetation cover intensity from full (A), to partial (E), to bare soil in (D).BC is a line depicting SMC levels from a wet surface (B) and semi-arid (D) to a completely dry surface in (C). BC is also called the soil line because it denotes the direction of drought severity.*F* is the line perpendicular to the soil line while dissecting the coordinate origin and parallel to the AD line.The *PDI* is the vertical distance from any given pixel point to line F, and the mathematical formula for the *PDI* is expressed through Equation (18) [26]:
(18)$$PDI=\frac{1}{\sqrt{M^{2}+1}}\left(R_{red}+MR_{NIR}\right).$$

The *PDI* can be a viable descriptor of SMC levels and distribution. In the NIR/red triangular feature space, points far from the normal line *F* represent dry surfaces and points near the same line correspond to wet surfaces [28]. The *PDI* is normalized, and it varies between 0 and 1, with 0 designated to low water stress and 1 to extreme water stress [26].

Since the *PDI* is dependent on NIR–red reflectance, any variability induced by the biophysical features of the soil, such as soil surface colour, vegetation species, and vegetation conditions, has a palpable effect on the index, so each study area requires a local calibration to determine its correspondent coefficient *M* (slope of the soil line) [26,29]. The *PDI* produces its best performance at low vegetation presence/bare soil applications. Conversely, in areas with variable vegetation cover intensities from bare soil to densely vegetated surfaces, it encounters a decrease in performance in terms of correlation, and it has inherent susceptibility to surfaces with non-flat topography, illumination factors, and cloud presence [28].

Despite the shortcomings discussed above, the *PDI* is still an effective indicator of SMC levels because of its strong correlation to SMC. However, in this paper, an additional index/model to counterbalance those limitations, along with the limitations imposed by the IEM inversion, was investigated. The selected model was the LST/VI triangular feature space or, more specifically, the TVDI.

### 2.4. Temperature Vegetation Dryness Index

The theoretical basis and biophysical properties of the *LST*/*VI* relationship have culminated in an index: the TVDI. This index operates under the assumption that the relationship between SMC levels, the intensity of fractional vegetation cover, and *LST* can be expressed through a two-dimensional scatter plot of which data points formulate a triangle/trapezoid shape [30,31]. Variations in SMC levels are modelled through plotting surface temperature as a function of fractional vegetation cover, with the latter often expressed by vegetation indices [32]. The difference in radiative temperatures between soil and vegetation canopy can be sensed via *L**ST*. Evapotranspiration is another factor that influences surface temperature through the energy balance at the surface [33]. The available energy for sensible heating of the surface increases whenever there is a decrease in evapotranspiration due to stomatal resistance to transpiration, which is an indication of soil moisture levels [34]. Therefore, the modelling of fractional vegetation cover and surface temperature facilitates the estimation of SMC for a different range of vegetation intensities [35]. The determination of the *TVDI* via the *LST*/VI triangular space provides surface SMC information, as depicted in Figure 3.

Figure 3 illustrates the concept of TVDI where:*LST* is the observed surface temperature (in Kelvin) at a random pixel.*LST_max_* is the regression line (least square) of the maximum surface temperatures observation for each of *VI* values, and *LST_max_* represents the dry edge denoted *LST_max_ = a*_1_ + *b*_1_*VI*.*LST_min_* is the regression line (least square) of the minimum surface temperatures observation for each of *VI* values, and *LST_min_* represents the wet edge denoted as *LST_min_* = *a*_2_ + *b*_2_*VI*.*a*_1_ and *b*_1_ are, respectively, the intercept and the slope of the linear dry edge (*LST_max_*), and *a*_2_ and *b*_2_ are, respectively, the intercept and the slope of the linear wet edge (*LST_min_*).The *TVDI* is expressed by Equation (19):
(19)$$TVDI=\frac{LST-LST_{min}}{LST_{max}-LST_{min}}$$
with the obvious meaning of variables.

*TVDI* values may range from 0 to 1, where 1 signifies the lowest levels of SMC and 0 indicates maximum evapotranspiration and water access (meaning high SMC levels). In [30], the authors compared *TVDI* values to soil moisture levels from a simulation produced by the MIKE SHE distributed hydrological model [36], determining that SMC and the *TVDI* have a relationship represented by a linear function (*m_v_* = x*TVDI* + y) that can be easily calculated using linear regression (R^2^ = 0.7). Subsequently, different variations of the *TVDI* were validated using in situ measurements in numerous investigations, yielding promising results [37,38].

The performance of the *TVDI* can be subject to a few sources of error, which can decrease the accuracy of its SMC retrieval, and those sources may cause few performance issues such as:The spatial and temporal variability of SMC is diminished because of the poor spatial and temporal resolution of the satellites pertinent to this model [38].The determination of the “triangle” from satellite data without huge data grids of large scale areas may become subjective—any given region of interest may not be spatially variable in terms of land surface conditions such as dry bare soil, wet bare soil, vegetation exhibiting water stress, and well-watered vegetation [30]. This can lead to difficulties in calculating an optimal dry and wet edge with generalisation capabilities due to local specific factors such as vegetation species, topography, net radiation, and cloud presence [31,39].Atmospheric effects and illumination effects (shadows) lead to the susceptibility of *LST* estimation errors, in addition to the fact that the *TVDI* only accounts for SMC in the top surface layer without any consideration of root zone SMC [35].

Despite the aforementioned limitations, the *TVDI* can still be considered a valid indicator of SMC levels, making it a viable addition to the fusion scheme of the previously discussed *IEM* inversion and the *PDI*. Its robustness for applications over large areas, as well as its insensitivity to surface cover type [30], can be beneficial for the proposed SMC estimation pipeline because it reduces any uncertainties brought about by the limitations of IEM inversion and the *PDI* in intensely vegetated areas. With this purpose, the authors of this paper opted to investigate whether the use of data fusion techniques, described in the next paragraph, could successfully ameliorate SMC retrieval accuracy.

### 2.5. SMC Estimation Scheme Fusion

In this paper, a novel soil moisture content estimation system is proposed, and the novelty of this system comes from the fusion aspect of the aforesaid estimations methods and indices at the feature and the decision levels. This system gains reconfigurable abilities via decision level fusion, which signifies that each method offers its independent estimation, and, consequently, in case of the absence of one data source, the system is still able to perform an estimation that is based on the other considered methods. Figure 4 illustrates a flowchart of the different components of such a system.

The components of this system are regrouped by functionality to facilitate their description; the groups are pre-processing, *PDI* and *TVDI* determinations, feature level fusion and fusion centre.

#### 2.5.1. Pre-Processing

All of the extracted EO data from Landsat-8 and Sentinel-1, respectively, were pre-processed using Sentinel Application Platform (SNAP) [40].

Initially, SAR data underwent radiometric calibration, then Range Doppler Terrain corrections, and finally multi-looking. Multispectral and thermal infrared datasets were transformed from digital numbers (DNs) to Top Of Atmosphere (TOA) reflectance and temperatures, respectively. Subsequently, the resulting SAR, multispectral, and thermal infrared products were resampled to 30 metres, which is both the spatial resolution of OLI and the sampling distance of the ground truth SMC measurements points. The 30 m resolution was a good compromise between the spatial resolution of TIRS (100 m) and the spatial resolution of Sentinel-1, which is (20.4 × 24.5 m). After image resampling, all products were co-registered. Then, the resampled co-registered MS image was used to calculate the *NDVI* and the *PDI* and to formulate the *LST/NDVI* feature space scatterplot when combined with the resampled co-registered thermal infrared image. On the other hand, the backscattering coefficient $\;\sigma^{0}$ from the SAR product, along with the RMS height (*s*), the calibrated parameter (*lopt*), and incidence angle *θ_i_*, were used to calculate the simulated backscattered coefficient, which was the product of EA-IEM.

#### 2.5.2. PDI and TVDI Determinations

For the *PDI*, the slope of the soil line was obtained via the least-squares linear regression of the co-registered resampled pixels of the red reflectance with the lowest NIR reflectance extracted from the MS images.

As for the *TVDI* determination, the NDVI was calculated through the resulting resampled MS images, and the LST was derived from the resampled thermal images. Then, the intercept and slope of the dry edge of the *LST/NDVI* feature space were calculated by selecting the maximum *LST* for each *NDVI* value and then applying least-squares linear regression to those temperatures. Conversely, to infer the intercept and slope of the wet edge, the minimum *LST* for each of the *NDVI* values were used instead.

#### 2.5.3. Feature Level Fusion

In this study, the feature level fusion was achieved through simple concatenations of the feature vectors extracted from the methods of SMC estimation discussed above.

Let *X = {x*_1_*, x*_2_ … *x_n_*} and *Y = {y*_1_*, y*_2_*, … y_m_}* denote feature vectors (*X*
∈
*R^n^* and *Y*
∈
*R^m^*). The idea is to merge vectors *X* and *Y* to generate a new joint feature vector *Z*, defined as [41]:(20)$$Z=X\cup Y=\{x_{1},x_{2}\;\ldots \;x_{n},y_{1},y_{2},\ldots y_{m}\},\;Z\;\in \;R^{n+m}$$

The study conducted in [42] concatenated features extracted from radar parameters {$\sigma_{VV}^{0}$*, θ_i_*}, vegetation index {*NDVI*}, and thermal image {*LST*}. The inclusion of the non-radar parameters seemed to increase the SMC estimation accuracy, as demonstrated by a lowered RMSE value in the order of 2.7% across all study areas. Similarly, the authors of [9] implemented a feature level fusion for SMC estimation by merging a feature vector containing radar and surface features {*s, l, θ_i_*, $\sigma_{VV}^{0}$} with the synergetic index {*TVDI*}. The implementation yielded less bias and smaller RMSE values by an order of 0.474%. However, in this study, three joint feature vectors with different combinations of features were used to test whether increasing the dimensionality of features space increased the overall accuracy of SMC estimations. The features were grouped into three joint feature vectors: FLF1, FLF2, and FLF3. Table 1 offers a description of the corresponding features composing each vector.

FLF1 is the same feature vector used in [9] with the replacement of *l* with *lopt*. The goal behind that specific selection of features was to introduce a new parameter resistant to vegetation cover presence (*TVDI*) to the EA-IEM inversion. FLF2 is composed of the *PDI* and the *TVDI* in cases when the surface roughness parameters are too high to be in the valid range for EA-IEM inversion. Furthermore, the multispectral component of this vector (*PDI*) enables the future exploitation of several potential multispectral imagers (such as Sentinel-2) if the temporal gap between the acquisition of Sentinel-1 and Landsat-8 data is too large. Finally, FLF3 is the joint feature vector composed of all of the available features. This was done to explore whether increasing the dimensionality of feature space even further resulted in a better SMC estimation in terms of accuracy.

The choice of the feature level fusion estimator has been the subject of the investigation of several studies [43,44,45,46,47]. The most popular estimation techniques were LS-regression, support vector machine (SVM), random forest (RF), and artificial neural network (ANN). Therefore, these methods underwent experimentation to ascertain which was the optimal estimator for this study. In this study, the ANN was the most accurate and consistent estimator in terms of performance. Its estimations were found to provide the strongest correlations along with the lowest RMSE values out of all other methodologies. Consequently, the authors of this study decided to perform all feature level fusion estimations through artificial neural networks.

An ANN is defined as a system that consists of artificial neurons interconnected by weights, and its basic structure is composed of an input layer, one (or more) hidden layer, and an output layer [48]. For this study, six different ANNs were created, with a different input vector assigned to each one. Furthermore, all ANN input vectors were trained, validated, and tested through the same ground truth SMC measurements.

Table 2 clarifies the input vectors for each ANN.

The backpropagation training algorithm for all ANNs is the Levenberg–Marquardt [48]. A total of 260 available samples were collected from all study areas: 70% of those samples were used for training, 20% were used for validation, and 10% were used for testing. There was an emphasis on the process of validation to ensure the adequate generalisation of the ANN as an attempt to minimise overfitting (when an ANN becomes too specific to a data sample) [49]. The size of the hidden layer size (10 nodes) and the specific division of the training sample was decided after several experiments from which it emerged that this specific configuration produced the most accurate results. Finally, all corresponding estimations were sent to the fusion centre, where a weight-based fusion was performed to improve the accuracy of SMC estimation.

#### 2.5.4. Fusion Centre

All estimations produced by the ANNs are the input arguments of the fusion centre function. The latter was an upgrade of a weight-based system designed by the author of this research in [50]. Moreover, instead of assigning weights w_1_, w_2_, and w_3_ to the estimations achieved by ANN_TVDI,_ ANN_PDI_, and ANN_EA-IEM_, respectively, the proposed weight-based system also assigns weights w_4_, w_5_, and w_6_ to estimations produced by ANN_FLF1_, ANN_FLF2_, and ANN_FLF3_, respectively. The reason behind this inclusion was to utilise the improved accuracy of the feature level fusion estimations to ameliorate the accuracy of the weight-based fusion estimation using Equation (21):(21)$${\mathrm{SMC}}_{\mathrm{fused}}=w_{1}{\mathrm{SMC}}_{\mathrm{TVDI}}+w_{2}{\mathrm{SMC}}_{\mathrm{PDI}}+w_{3}{\mathrm{SMC}}_{\mathrm{EA}-\mathrm{IEM}}+w_{4}{\mathrm{SMC}}_{\mathrm{FLF}1}+{\;w}_{5}{\mathrm{SMC}}_{\mathrm{FLF}2}+{\;w}_{6}{\mathrm{SMC}}_{\mathrm{FLF}3}$$
where SMC_fused_ is the weight-based decision level fusion estimation; SMC_TVDI_, SMC_PDI_, SMC_EA-IEM_*,* SMC_FLF1_, SMC_FLF2_, and SMC_FLF3_ are the output estimations of ANN_TVDI,_ ANN_PDI_, ANN_EA-IEM_, ANN_FLF1_, ANN_FLF2_, and ANN_FLF3_, respectively; and w_1_ + w_2_ + w_3_ + w_4_ + w_5_ + w_6_ = 1.

Figure 5 illustrates the inner workings of this fusion centre.

Initially, all weights are put in loops, where they are incremented from 0 to 1 with a step of 0.01. To optimise the process, each weight is incremented from 0 to 1 minus the value of the previous weights. Finally, SMC*_fused_* is only calculated if the sum of all of the weights is equal to 1. The metric selected for an estimation of the accuracy of estimation is the root mean square error (RMSE). After the RMSE of the final fusion is determined (RMSE*_fused_*), it is compared with the lowest achieved RMSE value out of all the estimations discussed above (RMSE_min_). If RMSE*_fused_* is lower than RMSE_min_, then RMSE*_fused_* becomes the new RMSE_min_.

Finally, after repeating this process for all combinations of weights satisfying the prescribed conditions, the weights respective to the lowest RMSE values (RMSE_min_) represent the optimal weights that are later used to derive the final fusion. Finally, said weights are saved for future SMC estimations.

## 3. Study Area

The validation and testing of the proposed methodology required direct measurements from suitable areas of interest or study areas. Therefore, three different study areas were selected to meet that requirement in this study.

The first dataset was relative to the agricultural fields in Blackwell farms, located in Guildford, a county town of Surrey in South East England. The size of the farm is approximately 295 × 308 m, and at the time of measurement collection, the field was spatially homogenous with sparse vegetation cover intensity. The field was also characterised as a non-flat surface topography (with just a little slope in the middle).

The second study area was an agricultural field in Sidi Rached, Tipasa, Algeria, and this region of interest is by far the largest (540 × 180 m). The field had a spatially heterogeneous cover intensity, with some areas containing minimal vegetation or bare soil, and other areas containing intense vegetation cover. However, this particular field was characterised by relatively flatter surface topography.

The third study area was another agricultural field in a different location at Sidi Rached, Tipasa. This field is the smallest of all of the datasets (180 × 180 m) due to restrictions made by the field owner. It was also visibly heterogeneous in terms of vegetation cover, and similarly to the second study, it presented a flat surface topography.

Table 3 contains detailed descriptions of all study areas, including location, coordinates, size, and the mineralogical composition of each soil (which is important for the EA-IEM inversion), as well as the respective NDVI means for the datasets as an indicator of the intensity of vegetation cover calculated from a TOA Landsat-8 product:

The selection of suitable study areas was predicated by the scarcity of concurrent acquisitions from different satellites. Indeed, Sentinel-1 and Landsat-8 acquisitions dates coincide on the same day only twice a month, and that was exacerbated by the frequently poor weather conditions in the United Kingdom. This complication prompted the authors to pursue different study areas elsewhere (in addition to the Blackwell farms datasets), and the chosen study areas were two agricultural fields in Sidi Rached, Tipasa, Algeria.

Figure 6 offers the point-of-view images of the considered agricultural fields.

The Blackwell farms dataset was characterised by low vegetation presence as visible in Figure 6a, whereas Sidi Rached 1 and 2 datasets were characterised by relatively thicker vegetation cover as illustrated by Figure 6b,c.

### 3.1. Earth Observation Data

Earth observation datasets were collected from two different satellites—Sentinel-1 as the active microwave sensor for the integral equation inversion model and Landsat-8 as the data source to calculate both the *PDI* and the *TVDI*.

#### 3.1.1. Sentinel-1

Sentinel-1 is a mission of twin satellites Sentinal-1A and Sentinel-1B, both equipped with a C-SAR on board (5.405 GHz frequency) [51].

All products used in this study were ground range detected (GRD), the acquisition mode was interferometric wide swath (IWS), and the available polarisations in this acquisition mode in the concerned datasets were VV and VH polarisations. However, due to the superior potential of co-polarized configurations at SMC estimation, only acquisitions with VV polarisation were considered [52]. The spatial resolution of this acquisition mode was 20 × 23 m, with a swath of 250 km. Table 4 provides further acquisition details:

#### 3.1.2. Landsat-8

The thermal infrared and multispectral data were generated from the Landsat-8 satellite. This satellite was chosen for its high spatial resolution in comparison to other satellites used in literature for similar purposes (such as MODIS) [53]. Landsat-8 has two sensors onboard: Operational Land Imager (OLI) and Thermal Infrared Sensor (TIRS). Landsat-8 data are acquired at 185 km swaths with a revisit time of 16 days [54]. Table 5 details the acquisition details for the study areas.

It is clear from inspecting Table 4 and Table 5 that there were temporal gaps between the times of acquisitions. It was unfortunately unavoidable due to the respective revisit cycles of the different used satellites in this research. The longest temporal gap occurred in the Blackwell farms datasets (approximately 20 h), while the other two datasets had a relatively shorter delay (approximately 5 h). Measures were taken to minimise the temporal gap; the ground truth measurement collection campaigns were scheduled in between the respective times of acquisitions of Landsat-8 and Sentinel-1. For Blackwell farms, the collection of SMC measurements was scheduled from 14:00 to 17:00. For Sidi Rached 1, SMC levels were measured from 11:30 p.m. to 14:00. For Sidi Rached 2, SMC measurements were collected from 09:00 (a.m.) to 10:30 (a.m.).

### 3.2. Ground Truth Measurements

There were two types of measurements to be collected for the proposed methodology: SMC levels and soil surface roughness. Three different instruments were used to collect such measurements: the ML3 theta soil moisture probe for SMC level measurements and the needle and laser profilometers for surface height measurements.

#### 3.2.1. ML3 Theta Soil Moisture Probe

The instrument used for SMC level measurements was the ML3 Theta probe soil moisture sensor, which was in a brand-new condition at the time.

When powering the ML3, it applies a 100 MHz waveform to an array of stainless-steel rods, which transmit an electromagnetic field to the soil. Any water content present in the soil surrounding those rods affects the permittivity *ɛ**_s_* of the soil (*ɛ**_w_* of water ≈ 81 and the *ɛ**_s_* of soil is ≈ 4). The ML3 derives the effect of the permittivity on the transmitted electromagnetic field in terms of stable voltage output, which represents a sensitive measure of SMC levels. The device has a measurement range of 0–100% with a 1% error for SMC values from 0 to 50% [55].

Since point-based SMC measurements can be described as less accurate depictions of the variability and distribution of SMC levels, four points of measurement were collected every 30 m. Subsequently, the mean of those four point-based measurements was assigned to its corresponding pixel of the earth observation data. The use of 30-m-distances between the points of measurements, in particular, had the purpose of matching the spatial resolution of the multispectral components of Landsat-8, which could be considered a good compromise between the spatial resolutions of thermal bands of Landsat-8 (100 m) and Sentinel-1 (20.4 × 22.5 m).

The process of SMC measurement took accurate logistical planning and scheduling to collect such measurement in a timely fashion. Not including the amount of time and effort necessary for the geo-location of the relevant data samples to be co-registered later with their corresponding earth observation pixels, about 2 h on average were required to collect around 100 data samples. Nevertheless, the collected measurements represented a good dataset for the objectives of this paper.

#### 3.2.2. Profilometer

The surface roughness profile (RMS height and correlation length) of the considered agricultural fields was also an important input parameter in the fusion block and needed to be determined. Two types of profilometers were used to collect such information at different stages of this study.

Initially, a needle profilometer was only used in the Blackwell farms field. To infer the measurements, the profilometer had to be positioned in the point of interest. Then, 78 needles (1 cm distance from each other) were inserted into this structure, and their height in the main structure was regulated accordingly to represent the height of the surface at each point. Once this process was done to all needles, the profilometer was superimposed on A0 paper where a curve of points representing the soil profile was drawn, as illustrated in Figure 7a.

The second device used to measure surface roughness parameters was a laser profilometer (depicted in Figure 7b). The device is composed of a metallic frame with a BOSCH PLR 15 Laser rangefinder fitted to it. The laser device has a range of 15 m, with a measuring error of 3 mm [56]. The distance between the equipped laser device and a flat surface was 0.33 m (which is well within its range). The laser pointed down to the soil surface, where the metallic structure was placed. Afterwards, a measurement was recorded, and the laser device was incrementally moved along a rail by one centimetre at a time through all 54 possible increments.

Table 6 delineates the measurements of different surface roughness parameters (*s* and *l*) expressed in cm, as collected for each study area. Blackwell farms were found to have a shorter correlation length than the other datasets, which could have been caused by the fact that the fields were recently ploughed at the time of the measurement campaign.

## 4. Results and Discussion

The results and analysis of the proposed SMC estimation system are organized into three different groups corresponding to each of the considered study areas: Blackwell farms, Sidi Rached 1, and Sidi Rached 2. The analysed estimations were named according to their input methods, as is visible in Figure 8, Figure 9 and Figure 10, where the measured SMC (%) is plotted as a function of estimated SMC (%) and the concerned methods of estimation are:TVDI.PDI.EA-IEM inversion.Feature Level Fusion 1 (FLF1) as the output of ANN_FLF1_.Feature Level Fusion 2 (FLF2) as the output of ANN_FLF2_.Feature Level Fusion 3 (FLF3) as the output of ANN_FLF3_.Weight-Based Fusion (WBF) as the output of the fusion centre.

### 4.1. Blackwell Farms

The first group of results are those pertinent to the Blackwell farms dataset. Figure 8 and Table 7 provide summaries of the results achieved by each estimation method.

Though soil moisture content values in this field were greater than 35% (the minimum measured SMC was 34.9%), the best estimation using a single method in terms of RMSE and degree of correlation was achieved via the *EA-IEM* inversion (RMSE = 1.7% and R = 0.57), which could be justified by the fact that the field had a non-flat surface and was mainly characterised by bare soil and sparse vegetation cover, which put the *PDI* in a disadvantage due to the nature of the surface topography and limited the performance of the *TVDI* because of the absence of the full range of vegetation cover (from bare soil to full vegetation).

What were also noticeable were that estimations using feature-level fusions consistently outperformed EA-IEM inversion in terms of RMSE and that those methods also had a stronger correlation. Furthermore, the addition of the synergetic feature *TVDI* to the feature vector of EA-IEM inversion in FLF1 caused immediate amelioration to the overall accuracy of estimation (RMSE = 1.54%) and correlation (R = 0.66). This addition balanced out some of the inaccuracies that were caused by the high SMC values and surface roughness of the site. As for FLF2, the elimination of radar and surface roughness features, as well as the addition of a feature (*PDI*) insensitive to the narrow range of vegetation cover of the study area, slightly improved the accuracy and increased the correlation of this estimation (RMSE = 1.53% and R = 0.67). Moreover, the inclusion of all of the available features in FLF3 produced the most accurate feature level fusion estimation in terms of RMSE and correlation (RMSE = 1.37% and R = 0.75), which was quite a discernible improvement. Finally, the output of WBF produced the best accuracy and the strongest correlation out of all of the used methods (RMSE = 1.32% and R = 0.77), and the weights achieved for this study area were:(22)$${\mathrm{SMC}}_{\mathrm{WBF}}=0.08{\mathrm{SMC}}_{\mathrm{TVDI}}+0.26{\mathrm{SMC}}_{\mathrm{FLF}2}+0.66{\mathrm{SMC}}_{\mathrm{FLF}3}$$

The WBF method disregarded the *PDI, EA-IEM* inversion, and FLF1 estimations completely and assigned the largest importance to FLF3 (weight = 0.66), which was reasonable since it was the most accurate feature level estimation. Then, the second-largest importance was for FLF2 (weight = 0.26), which was also the second-best estimation in terms of RMSE values. Finally, some importance was assigned to the *TVDI* (weight = 0.08).

For this study area, the WBF method produced the best results in terms of RMSE and R, with FLF3 in a very close second. However, the latter does possess a slightly more accurate range of values in terms of minimum, maximum, mean, and standard deviation.

Figure 9 depicts SMC maps of the produced SMC estimation through the proposed system, as well as of field measurements:

By observing Figure 9, it can be concluded that there was a great resemblance between the estimated SMC map and the measured SMC map, especially at higher values (SMC > 40%). However, the proposed estimation method produced its poorest performance, especially in the 37%–40% interval and, similarly but to a lesser extent, in the 42–44% interval, as visible in Figure 8g and Figure 9a.

### 4.2. Sidi Rached 1

The second group of results are those corresponding to the Sidi Rached 1 dataset, and Figure 10 and Table 8 provide summaries of the results achieved by each estimation method.

Table 8 showcases higher RMSE values that could be attributed to higher spatial variability in terms of vegetation cover intensity and soil moisture content. This assumption was justified by the fact that these agricultural fields had a higher standard deviation in terms of SMC measurements (SD = 4.94%). The most accurate estimation using a single method in terms of RMSE and degree of correlation was yet again achieved through the *EA-IEM* inversion (RMSE = 4.1% and R = 0.54). The relatively lower performance of the *TVDI* could be attributed to the low number of pixels representing bare soil and low vegetation intensity, as only 24% of all pixels had *NDVI* values less than 0.3, which consequently led to a less accurate determination of the dry edge. Meanwhile, the performance of the *PDI* could also be considered subjective due to the high number of pixels corresponding to dense vegetation, the soil line was therefore not accurately depicted when considering that 76% of the pixels had *NDVI* values of more than 0.3.

It was observable that feature-level fusion estimations also outperformed *EA-IEM* inversion in terms of RMSE and strength of correlation in this study area. Indeed, the inclusion of the *TVDI* to the feature vector of EA-IEM inversion in FLF1 made a positive impact on the accuracy of estimation (RMSE = 3.17%), as well as the correlation (R = 0.76). Additionally, that addition lowered the effect of the dense vegetation cover on the accuracy of EA-IEM inversion estimation and especially increased the accuracy of estimation for values ranging from 28% to 33% (as visible in Figure 10d). As for the FLF2 estimation, dissimilar to the results achieved in the first study area, the exclusion of radar and surface roughness features did not yield a more accurate estimation (RMSE = 3.89% and R = 0.62), which could have been due to the same reasons influencing the estimations of the *PDI* and the *TVDI* individually. For FLF3, the inclusion of all of the available features once again produced the best feature level fusion estimation in terms of accuracy and correlation (RMSE = 2.9% and R = 0.81).

Finally, the estimation produced by WBF was characterised by the best accuracy and the strongest correlation out of all of the considered estimation methods (RMSE = 2.7% and R = 0.84), and the weights achieved for this study area were:(23)$${\mathrm{SMC}}_{\mathrm{WBF}}=0.36{\mathrm{SMC}}_{\mathrm{FLF}1}+0.08{\mathrm{SMC}}_{\mathrm{FLF}2}+0.56{\mathrm{SMC}}_{\mathrm{FLF}3}$$

The WBF method completely dismissed the *TVDI*, the *PDI,* and *EA-IEM* inversion estimations. Indeed, it seemed to assign weights according to the accuracy of each estimation, with the largest weight assigned to the most accurate feature-level estimation FLF3 (weight = 0.56), the second-largest weight to estimation FLF1 (weight = 0.36), and the smallest weight to *FLF2* (weight = 0.08).

Additionally, for this study area, the WBF method produced the best results in terms of RMSE and R, as well as the range of values in terms of minimum, maximum, mean, and standard deviation. Furthermore, the correlation of WBF estimation in Sidi Rached 1 (R = 0.84) was stronger than that of the Blackwell farms counterpart (R = 0.77), which could be attributed to the shorter time gap between different satellite acquisitions in this area compared to the Blackwell farms dataset.

Figure 11 depicts SMC maps of the produced SMC estimation through the proposed system, as well as of field measurements.

Figure 11 once again exhibits a large agreement between the SMC estimation with the proposed method and the measured SMC map. The highest degree of similarities could be observed at 30–40% interval. However, in this study area, the system did seem to struggle with lower SMC values, especially values of less than 22%, as visible in Figure 10g and Figure 11a.

### 4.3. Sidi Rached 2

The third group of results are those relative to the Sidi Rached 2 dataset, for which the smallest number of in situ SMC measurements (*n* = 36) was available. Figure 12 and Table 9 offer detailed descriptions of the results achieved by each estimation method.

The most accurate estimation using a single method in terms of RMSE and degree of correlation was obtained through the *TVDI* method this time (RMSE = 2.32% and R = 0.69). This result could have been influenced by the presence of a full range of vegetation cover intensity, manifested in 41% of the overall pixels with *NDVI* values below 0.3. The performance of the *PDI* was slightly poorer, which was coherent with the presence of pixels corresponding to dense vegetation that, in turn, may have had an adverse impact on the determination of the soil line. Another viable reason for the *TVDI* and the *PDI* outperforming EA-IEM inversion could have been that the time of ground truth collection (from 09:00 a.m. to 10:30 a.m.) was relatively closer to the Landsat-8 acquisition time (10:25 a.m.) than to the Sentinel-1 (05:45 a.m.).

The feature-level fusion-based estimations also outperformed the *TVDI* estimation in terms of RMSE and degree of correlation in this study area. FLF1 and FLF2 performed almost identically by exhibiting the same degree of correlation (R = 0.79), with an insignificant difference in RMSE values (1.97% and 1.98%, respectively). Regarding FLF3, the inclusion of all the available features generated the best feature level fusion estimation in terms of accuracy and correlation (RMSE = 1.34% and R = 0.91) for this dataset as well.

Finally, the WBF estimation once more produced the best accuracy and the strongest correlation out of all of the considered methods (RMSE = 1.27% and R = 0.93), and the weights obtained for this study area were:(24)$${\mathrm{SMC}}_{\mathrm{WBF}}=0.04{\mathrm{SMC}}_{\mathrm{TVDI}}+0.14{\;\mathrm{SMC}}_{\mathrm{PDI}}+0.08{\mathrm{SMC}}_{\mathrm{FLF}1}+0.02{\mathrm{SMC}}_{\mathrm{FLF}2}+0.72{\mathrm{SMC}}_{\mathrm{FLF}3}$$

The WBF method completely ignored the contribution of the *EA-IEM* inversion estimation, and, this time, it generated an interesting configuration of weights by assigning the largest significance to the most accurate feature level estimation FLF3 (weight = 0.72), the second-largest weight to *PDI* estimation (weight = 0.14), and then smaller weights to the *FLF1* estimation (weight = 0.08), the *TVDI* estimation (weight = 0.04), and the FLF2 estimation (weight = 0.02). Eventually, in Sidi Rached 2, the WBF method produced the most accurate results in terms of RMSE and R, as well as the closest to the measured SMC in terms of the range of the values of minimum, maximum, and mean. The strong correlation of WBF estimation in this study area (R = 0.93) supported the assumption that the correlation could be ascribed to the length of the temporal gap between the times of acquisitions.

Figure 13 depicts SMC maps of the produced SMC estimation through the proposed system, as well as of field measurements:

Figure 13 validates the findings of the previously detailed study areas. The product of the proposed SMC system in this study area was almost identical to the measured SMC map. However, this degree of similarity could very well have been due to the low number of data samples.

### 4.4. Remarks and Discussions

The results of the proposed approach were in good agreement with the findings of [9,50]. The inversion of the EA-IEM produced the lowest RMSE out of all estimations using an individual method in Blackwell farms and Sidi Rached 1, while the TVDI produced the most accurate one in Sidi Rached 2. This result is logical for the Blackwell farms dataset, but it is not the case for Sidi Rached 1 since it was characterised by an intense vegetation cover. This could be explained by the absence of the full range of vegetation cover for Sidi Rached 1, especially if one examines the performance of the TVDI in the Sidi Rached 2 dataset. Out of all estimations produced by feature level fusions, FLF3 generated the most accurate estimation across all datasets. This seemed to validate the assumption that maximising the dimensionality of the features space increased the accuracy of estimation. Indeed, regarding FLF1 and FLF2 estimations, they performed almost identically in Blackwell farms and Sidi Rached 2, while FLF1 did outperform FLF2 in Sidi Rached 1, which supported the assumption that the absence of a full range of vegetation cover had a negative impact on the accuracy of SMC estimation through the TVDI. Finally, the proposed fusion centre systematically and consistently improved the accuracy of SMC estimation manifested in lower RMSE values in the orders of at least 0.38%, 1.4%, and 1.09% for Blackwell farms, Sidi Rached 1, and Sidi Rached 2 datasets, respectively, which were good improvements on the accuracy of estimation obtained by a single method. However, more investigations are necessary to understand the meaning of weight assignments.

The performance of the proposed SMC estimation system was sensitive to the vegetation cover intensity, whether be it its degree or its distribution. Furthermore, the proposed system produced its optimal performance in bare soil or low vegetation cover (Blackwell farms). It also produced a good performance in datasets with a full range of vegetation cover intensity from bare soil to dense vegetation, as in the case of Sidi Rached 2, but its performance was relatively poorer in fully vegetated areas (Sidi Rached 1). This could have been due to the fact that the TVDI was the only feature that accounted for the degree of vegetation intensity implicitly through NDVI. However, even the TVDI had its issues—though the stomatal aperture increased with the increase of temperature, it decreased or even came close to avoiding dehydration when temperatures surpassed a certain threshold, which may have generated dry/wet boundaries inconsistent with the theoretical edges of *LST*/NDVI feature space [57]. Another TVDI-related issue is that a full range of vegetation is a key requirement for the objective determinations of its dry and wet edges, which was not the case in any of the considered study areas. A possible solution for this is the consideration of parabolic dry and wet edges instead of classical oblique lines [57]. Another solution could be the consideration of theoretical methods to determine the dry/wet edges. These methods consist of theoretical calculations that include physical processes, which means that the determination of the boundaries is calculated via an energy balance equation of the land surface instead of the used linear fitting, which would lead to more general boundaries [58]. Regarding IEM, it was quite sensitive to vegetation cover. Exponential or Gaussian correlation lengths have been suggested in the literature, as also reported in this paper, as models to adopt in cases of low or high roughness. Fewer experiments regarding which correlation function to adopt in case of inhomogeneity and areas presenting a variety of surface roughnesses are available. Forcing one function or another to the entire area is the simplistic approach that is often used, but, with the increased spatial resolution of spaceborne sensors, it will become possible to apply different functions within the same area. In the case of homogeneity, i.e., in presence of soil surfaces with comparable surface conditions, we already know from [59] that experimental measurements of backscattering coefficients are much closer to the predictions of the IEM for a surface with an exponential correlation than a Gaussian correlation.

Besides the performance issues described above, the authors of this study are conscious of the small number of the available data samples for ANN training, which could very well have caused overfitting issues. Overfitting could limit the constraints of applicability of the proposed SMC estimation system. Overfitting could potentially lead the proposed system to become specialised for areas with similar characteristics to those of the study areas in terms of SMC distribution, vegetation cover intensity, and vegetation cover distribution, as well as soil composition and surface roughness parameters. However, this could be corrected by the consideration of data from the International Soil Moisture Network [60] or data from the cosmic-ray soil moisture monitoring network (Cosmos-UK) [61]. Additionally, there is always the possibility of undertaking additional field campaigns for the collection of additional ground truth measurements. Considering new and upcoming constellations of satellite data, the temporal resolution of the considered satellite data acquisition is another area of certain future improvement. A promising example is a Landsat-9 mission (which is scheduled to be launched in September 2021), which predominantly replicates its predecessor Landsat-8 in terms of onboard sensors [62]. Landsat-9 in conjunction with Landsat-8 will offer a revisit frequency of eight days [63]. Consequently, the improved temporal coverage of multispectral and thermal infrared data will increase the accuracy and correlation of the collected measurements with the earth observation data.

## 5. Conclusions

The results of a novel SMC estimation system were evaluated and analysed according to each study area. The proposed system produced its lowest RMSE values in the Sidi Rached 2 dataset (1.27%), followed by the Blackwell farms dataset (1.34%) and, finally, by the Sidi Rached 1 dataset (2.7%). Based on the analysed areas, it is safe to conclude that the proposed system was more accurate than any of the estimations produced by a single method. It is also safe to conclude that the system performed best in presence of a full range of vegetation cover. Notwithstanding these promising results, there were some other concerns relating to the performance of the proposed system. The small number of collected SMC measurements could have very possibly caused overfitting. Another concern is the temporal gap between the acquisition times of the considered satellites, which may have adversely impacted the correlation of EO-data to SMC levels. The revisit time of the satellites could be also considered as an additional limitation, especially in the case of Landsat-8 (16 days), which is particularly important if SMC level monitoring is relevant to agricultural practices. The impact of the latter limitation could be diminished by the inclusion of additional satellites such as Sentinel-2 and the likely addition of Landsat-9 data after its launch. Finally, further research is required to address the performance issues of the system in the absence of a full range of vegetation cover.

## Figures and Tables

**Figure 1 sensors-21-03457-f001:**
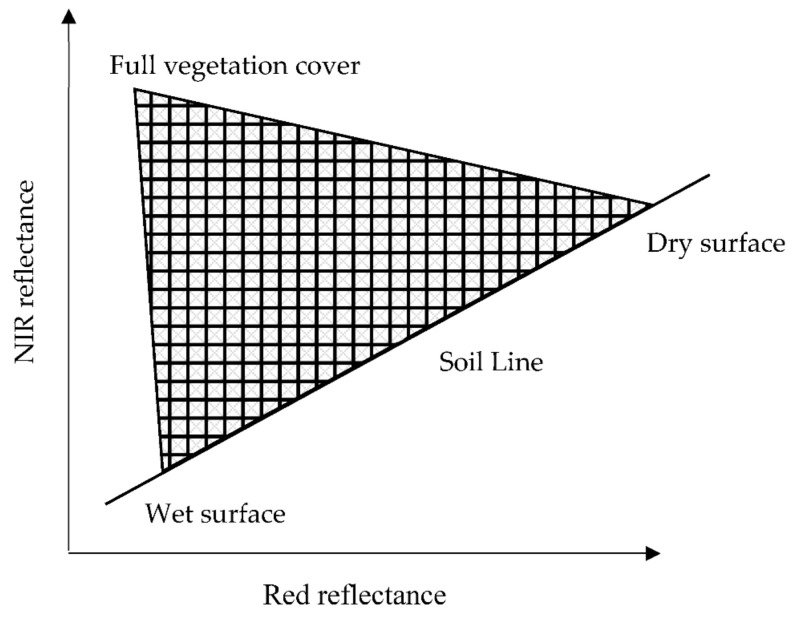
NIR/red triangular feature space. Adapted from [27]. Reprinted with permission from ref. [27]. Copyright 2011 Elsevier Ltd.

**Figure 2 sensors-21-03457-f002:**
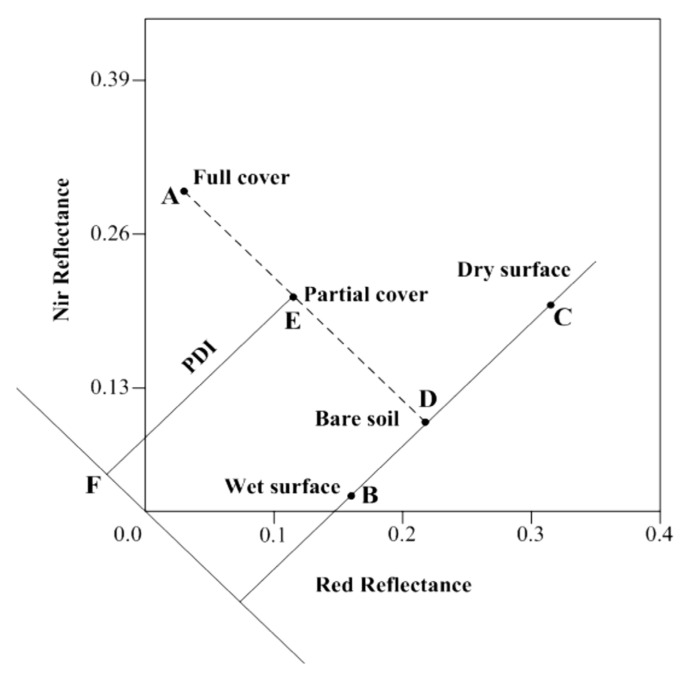
Definition of the *PDI.* Adapted from [26]. Reprinted with permission from ref. [26]. Copyright 2006, Springer-Verlag.

**Figure 3 sensors-21-03457-f003:**
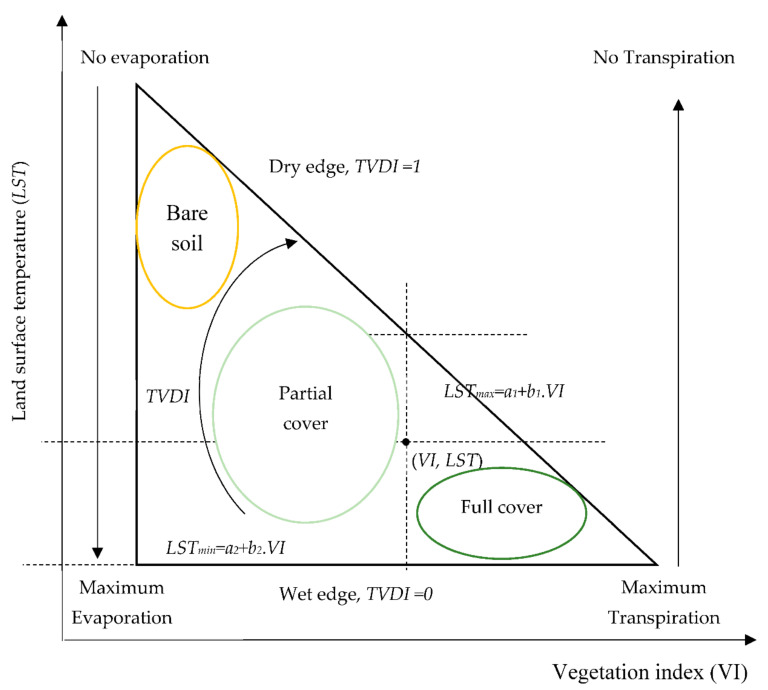
Definition of the *TVDI* in the *LST/VI* feature space. Adapted from [30,32]. 1. Reprinted with permission from ref. [30]. Copyright © 2001 Elsevier Science Inc. 2. Reprinted with permission from ref. [32]. Rights managed by Taylor & Francis.

**Figure 4 sensors-21-03457-f004:**
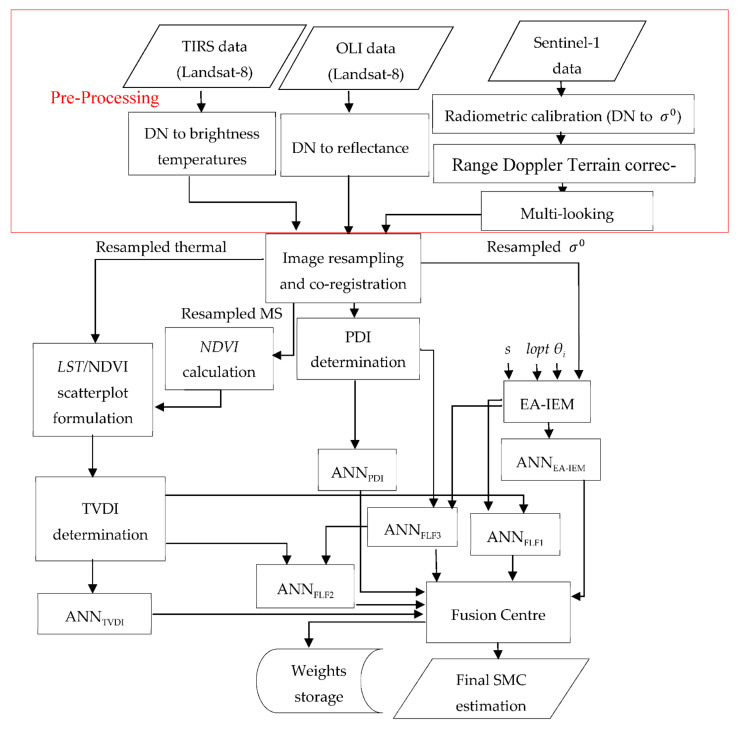
The proposed soil moisture content estimation system.

**Figure 5 sensors-21-03457-f005:**
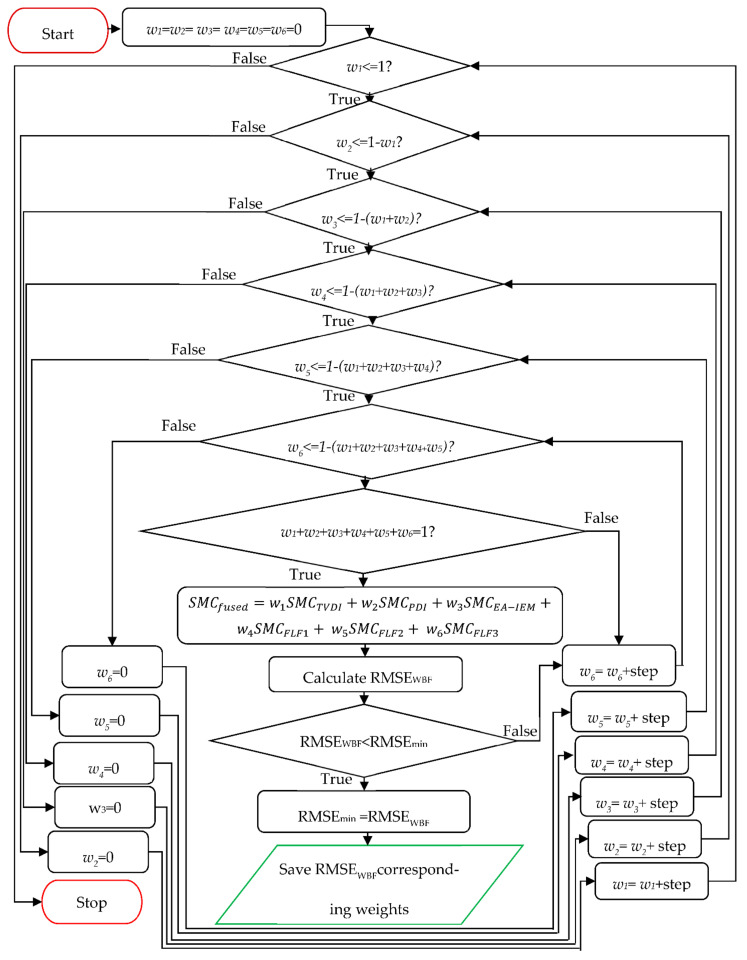
Flowchart of the fusion centre.

**Figure 6 sensors-21-03457-f006:**
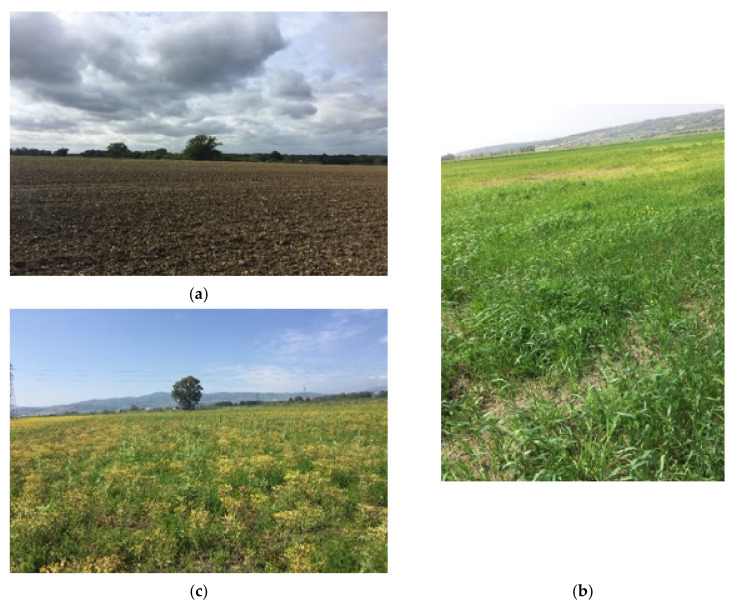
Field photographs of the landscapes of Blackwell farms (**a**), Sidi Rached 1 (**b**), and Sidi Rached 2 (**c**).

**Figure 7 sensors-21-03457-f007:**
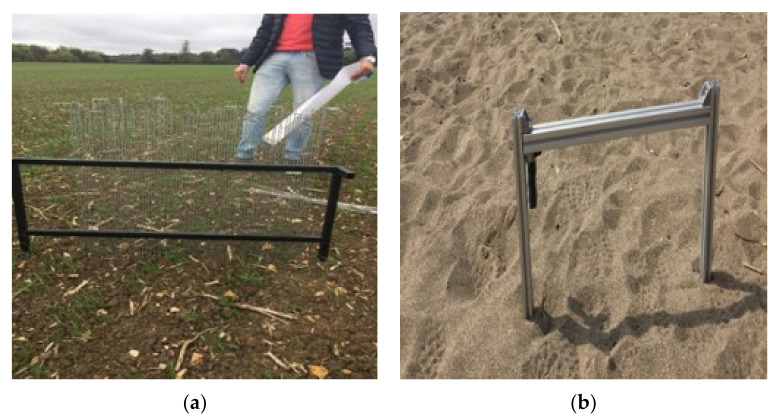
The process of surface roughness parameters measurement using a needle profilometer (**a**) and a laser profilometer (**b**).

**Figure 8 sensors-21-03457-f008:**
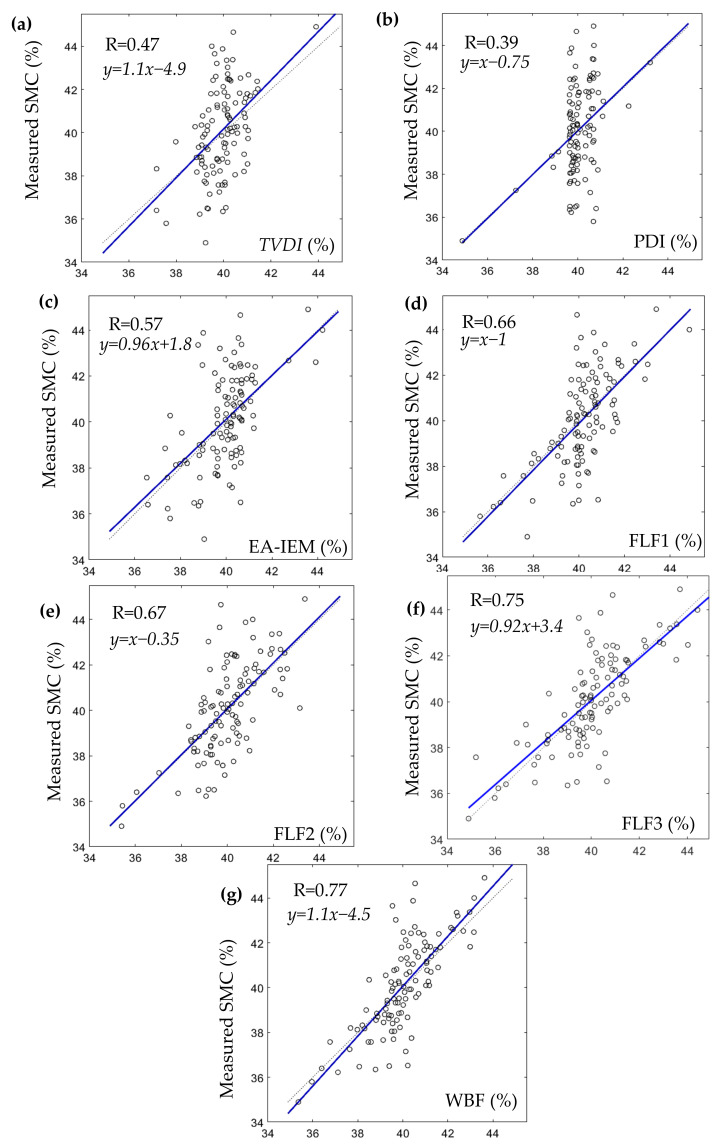
Results of each of the used SMC estimation methods in Blackwell farms datasets where: (**a**). TVDI. (**b**). PDI. (**c**). EA-IEM. (**d**). FLF1. (**e**). FLF2. (**f**). FLF3. (**g**). WBF.

**Figure 9 sensors-21-03457-f009:**
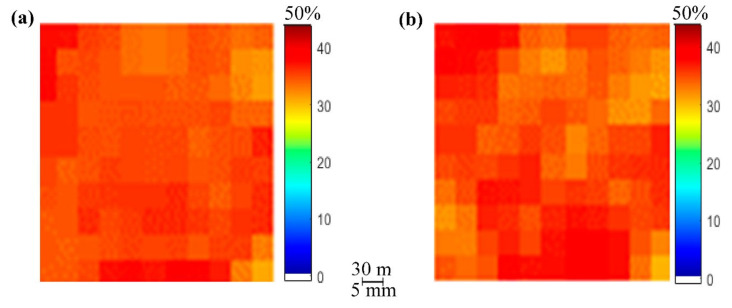
SMC maps: (**a**) produced by the estimation using the weight-based methodology and (**b**) produced by field measurements (17 November 2017).

**Figure 10 sensors-21-03457-f010:**
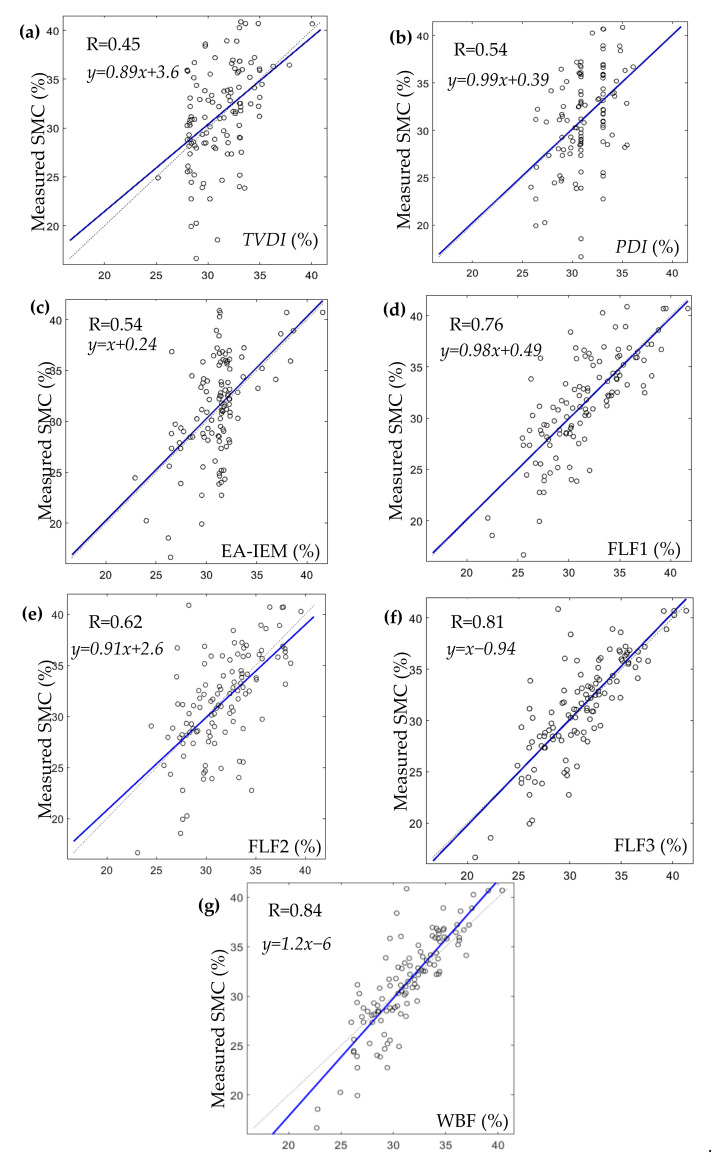
Results of each of the used SMC estimation methods in the Sidi Rached 1 dataset where (**a**). TVDI. (**b**). PDI. (**c**). EA-IEM. (**d**). FLF1. (**e**). FLF2. (**f**). FLF3. (**g**). WBF.

**Figure 11 sensors-21-03457-f011:**
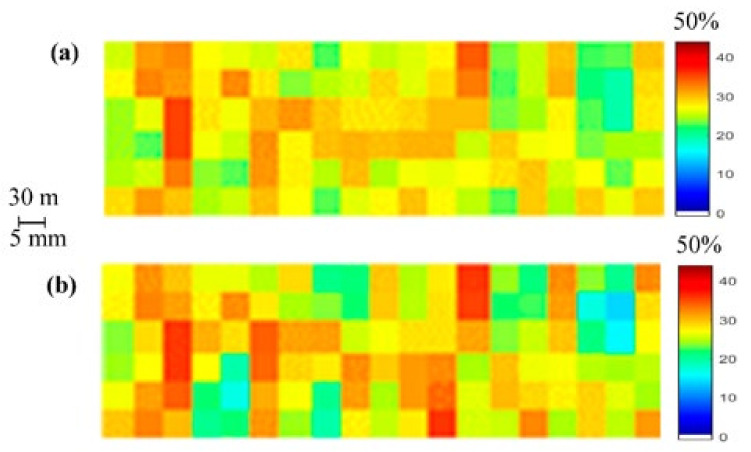
SMC maps: (**a**) produced by the estimation using the weight-based methodology and (**b**) produced by field measurements (7 April 2018).

**Figure 12 sensors-21-03457-f012:**
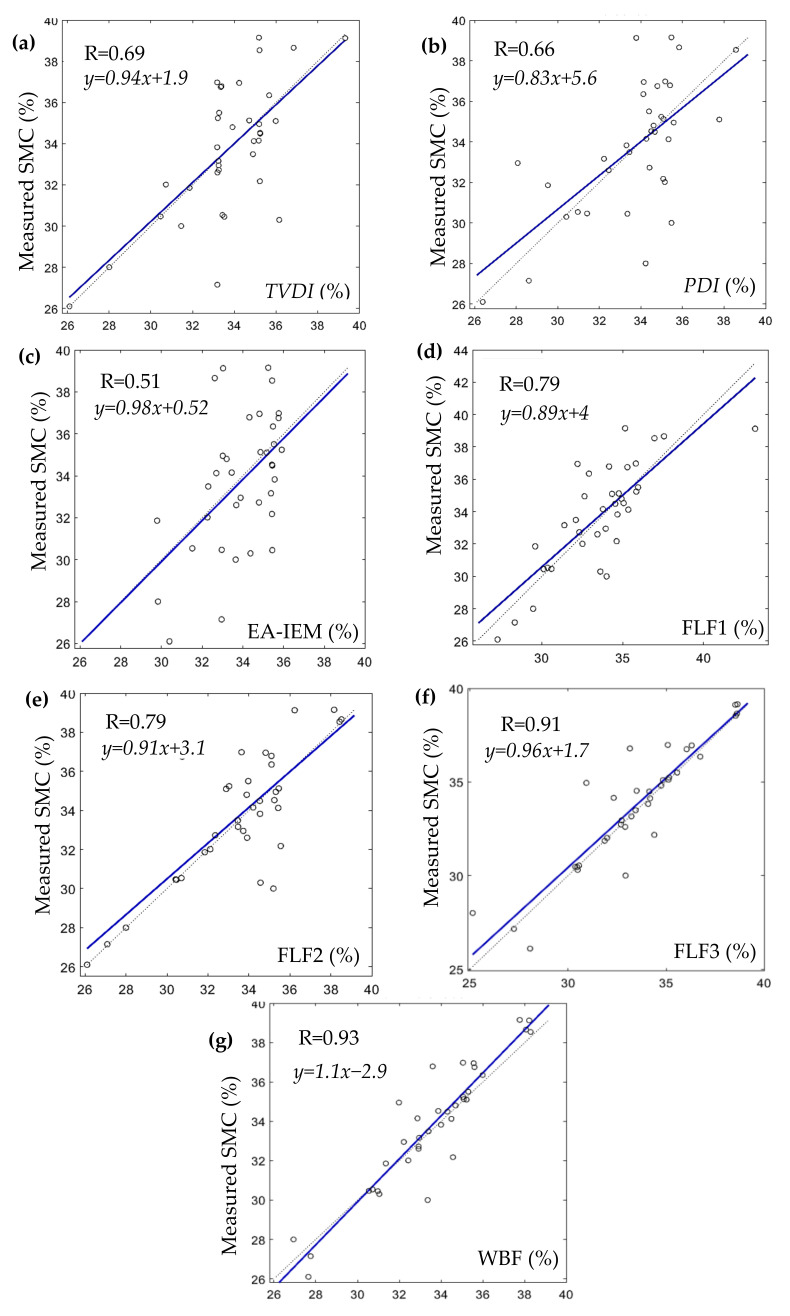
Results of each of the used SMC estimation methods in the Sidi Rached 2 dataset where: (**a**). TVDI. (**b**). PDI. (**c**). EA-IEM. (**d**). FLF1. (**e**). FLF2. (**f**). FLF3. (**g**). WBF.

**Figure 13 sensors-21-03457-f013:**
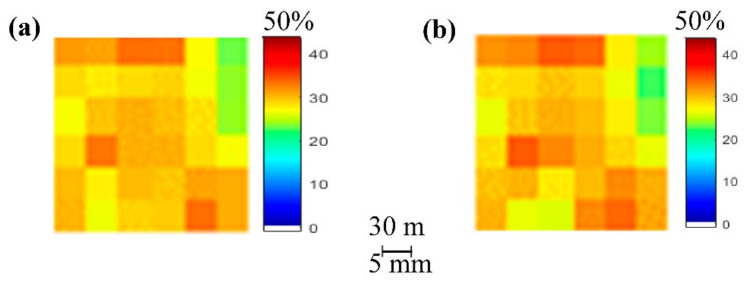
SMC maps: (**a**) produced by the estimation using the weight-based methodology and (**b**) produced by field measurements (9 May 2018).

**Table 1 sensors-21-03457-t001:** Feature level fusion joint vectors.

Joint Feature Vector	Features Composition
FLF1	$\{s,\mathrm{lopt},\theta_{i}$$,\sigma_{VV}^{0}$} *∪ {TVDI*}
FLF2	{*TVDI} ∪ {PDI*}
FLF3	$\{s,\mathrm{lopt},\theta_{i}$$,\sigma_{VV}^{0}$} *∪* {*TVDI} ∪ {PDI*}

**Table 2 sensors-21-03457-t002:** Description of the input feature vectors of all of the used artificial neural networks.

ANN	Input Feature Vector
ANN_TVDI_	*TVDI*
ANN_PDI_	*PDI*
ANN_EA-IEM_	$(s,\mathrm{lopt},\theta_{i}$$,\sigma_{VV}^{0}$)
ANN_FLF1_	$(s,\mathrm{lopt},\theta_{i}$$,\sigma_{VV}^{0}$,*TVDI*)
ANN_FLF2_	(*TVDI*,*PDI*)
ANN_FLF3_	$(s,\mathrm{lopt},\theta_{i}$$,\sigma_{VV}^{0}$,*TVDI,PDI*)

**Table 3 sensors-21-03457-t003:** A detailed description of the used study areas.

Study Area	Location	Coordinates (Latitude, Longitude)	Size (m × m)	Soil Type	NDVI (Mean)
Blackwell farms	Guildford, Surrey, United Kingdom	51°14′10′′ N, 000°37′32′′ W	295 × 308	Clay loam	0.26
Sidi Rached 1	Tipasa, Algeria	36°33′ 18′′ N, 002°31′28′′ E	540 × 180	Sandy loam	0.43
Sidi Rached 2	Tipasa, Algeria	36°31′30′′ N, 002°32′38′′ E	180 × 180	Sandy loam	0.33

**Table 4 sensors-21-03457-t004:** Sentinel-1 acquisition details.

Study Area	Spatial Resolution (Range, Azimuth)	Incidence Angle (°) (Min–Max)	Acquisition Date	Acquisition Time
Blackwell farms	20.4 × 22.5 m	38.2–41.52	18 November 2017	06:21
Sidi Rached 1		44.98–45	7 April 2018	17:51
Sidi Rached 2		34.4–34.41	9 May 2018	5:45

**Table 5 sensors-21-03457-t005:** Landsat-8 acquisition details.

Study Area	Acquisition Date	Acquisition Time
Blackwell farms	17 November 2017	10:52
Sidi Rached 1	7 April 2018	10:25
Sidi Rached 2	9 May 2018	10:25

**Table 6 sensors-21-03457-t006:** The values of the measured surface roughness parameters.

Study Area	Profilometer Type	*s* [cm]	*l* [cm]
Blackwell Farms	Needle	1.57	1.67
Sidi Rached 1	Laser	2.85	5.08
Sidi Rached 2		1.76	8.66

**Table 7 sensors-21-03457-t007:** The results of each of the used estimation methods in the Blackwell farms dataset.

Estimation Methods		Blackwell Farms (*n* = 110)								
		Estimated SMC					Measured SMC			
		RMSE (%)	Min (%)	Max (%)	Mean (%)	SD (%)	Min (%)	Max (%)	Mean (%)	SD (%)
Estimations using a single method	*TVDI*	1.82	37.16	43.9	39.91	0.87	34.9	44.9	40.07	2.07
	*PDI*	1.9	34.88	43.2	40.04	0.8				
	*EA-IEM*	1.7	36.52	44.21	39.96	1.23				
Estimations using feature-level fusion	*FLF1*	1.54	35.65	44.85	40.18	1.34				
	*FLF2*	1.53	35.39	43.36	40.01	1.37				
	*FLF3*	1.37	34.88	44.42	40.02	1.69				
Estimation using decision level fusion	*WBF*	1.32	35.36	43.62	40.01	1.43				

**Table 8 sensors-21-03457-t008:** The results of each of the used estimation methods in the Sidi Rached 1 dataset.

Estimation Methods		Sidi Rached 1 (*n* = 114)								
		Estimated SMC					Measured SMC			
		RMSE (%)	Min (%)	Max (%)	Mean (%)	SD (%)	Min (%)	Max (%)	Mean (%)	SD (%)
Estimations using a single method	*TVDI*	4.41	25.16	40.07	31.14	2.52	16.65	40.87	31.4	4.94
	*PDI*	4.4	25.85	36.08	31.25	2.26				
	*EA-IEM*	4.1	22.91	41.55	31.12	2.67				
Estimations using feature-level fusion	*FLF1*	3.19	22.05	41.68	31.48	3.84				
	*FLF2*	3.89	23.1	39.55	31.67	3.38				
	*FLF3*	2.9	20.75	41.32	31.22	3.87				
Estimation using decision level fusion	*WBF*	2.7	22.67	40.4	31.35	3.51				

**Table 9 sensors-21-03457-t009:** The results of each of the used estimation methods in the Sidi Rached 2 dataset.

Estimation Methods		Sidi Rached 2 (*n* = 36)								
		Estimated SMC					Measured SMC			
		RMSE (%)	Min (%)	Max (%)	Mean (%)	SD (%)	Min (%)	Max (%)	Mean (%)	SD (%)
Estimations using a single method	*TVDI*	2.32	26.09	39.32	33.73	2.34	26.1	39.15	33.75	3.2
	*PDI*	2.44	36.83	38.57	33.69	2.53				
	*EA-IEM*	2.74	29.78	35.9	33.92	1.69				
Estimations using feature-level fusion	*FLF1*	1.97	27.26	43.19	33.58	2.87				
	*FLF2*	1.98	26.09	38.51	33.56	2.77				
	*FLF3*	1.34	25.14	38.62	33.46	3.05				
Estimation using decision level fusion	*WBF*	1.23	26.93	38.27	33.51	2.72				

## Data Availability

The Sentinel-1 SAR product presented in this study is openly available at the Copernicus Open Access Hub. The Landsat-8 product is openly available at https://earthexplorer.usgs.gov/ (accessed on 15 May 2021).

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
