# Peer review of "Novel Weight-Based Approach for Soil Moisture Content Estimation via Synthetic Aperture Radar, Multispectral and Thermal Infrared Data Fusion"

_sensors, 2021, doi:10.3390/s21103457_

Round 1

Reviewer 1 Report

SMC estimations using the High-resolution SAR, exhibits a questionable performance when the areas of interest are characterized by medium to intense vegetation covers, especially at very short radar wavelength. However, SMC estimations using multispec-tral and thermal infrared synergies are not the least affected by the presence of partial or even full vegetation covers. Therefore, the data fusion techniques can be seen as a potential solution. This paper an Artificial Neural Network (ANN)-based data fusion method. In my opinion, the proposed method is technically correct and experiments are also used to verify. Some minor revisions should be considered.

1: In the proposed method, three feature level fusions are performed. The accuracy of each extracted feature is important. As the authors says, different methods have different results under different scenes. Please give a short discussion.

  1. The relationship of the dielectric constant and SMC has been described as a polynomial. Can we explore the noise characteristic or use the CS-based method to get more accurate estimation?

[1] J. Zheng, R. Chen, T. Yang, X. Liu, H. Liu, T. Su and L. Wan, “An efficient strategy for accurate detection and localization of UAV swarms,” IEEE Internet of Things Journal, DOI 10.1109/JIOT.2021.3064376, 2021.

[2] Fiscante N, Addabbo P, Clemente C, et al. A track-before-detect strategy based on sparse data processing for air surveillance radar applications[J]. Remote Sensing, 2021, 13(4): 662.

[3] J. Zheng, T. Yang, H. Liu, T. Su and L. Wan, “Accurate detection and localization of UAV swarms-enabled MEC system,” IEEE Transactions on Industrial Informatics, , vol. 17, no. 7, pp. 5059-5067, 2021.

Reviewer 2 Report

Remote sensing SAR technologies exhibit potential estimations for Soil Moisture Content. Such technologies do also present several performance disadvantages. The authors in this study propose a novel data fusion methodology for SMC estimation. Three methods: the inversion of an Empirically Adapted Integral Equation Model via SAR data, the Perpendicular Drought Index and the Temperature Vegetation Dryness Index from Landsat-8 data. Three feature level fusions are performed producing three different novel features combinations where features go to an ANN. All SMC estimations are fused at the decision level. The performance of the designed system has been validated in experiments from three study areas, an agricultural. The SMC estimation system appears to demonstrate stronger correlations and lower RMSE values in comparison with state-of-the art methods.

The paper has been well written, explaining different aspects connected with the problem. Some small correction should be done before publishing this paper.

Comments:

  1. The performance of your SMC estimation system is sensitive to the vegetation cover. Also, you noticed that the experimental results presented better performance for bare soil or low vegetation. Please explain this validation results with much more details and maybe propose future modification of your method that can increase the quality.
  2. There are serious problem that follows from the small number of the available data samples for ANN training. This problem is common for ANN training. How do think to increase data in training stage maybe generating novel data?
